# Home-based nurturing care practices for children under five with low socioeconomic position in Santo Domingo, Dominican Republic

**Adrianne Katrina Nelson**[1]\*, **Laura Sánchez-Vincitore**[2], **Melanie Patricia Frias**[2], **Michelle Marie Susana**[2], **Carl Kendall**[3,4], **Katherine Theall**[4], **Martha Vibbert**[5,6], **Heidi Luft**[7], **Arachu Castro**[8]

**1** Department of Health Policy and Management, Yale School of Public Health, New Haven, Connecticut, United States of America, **2** Laboratorio de Neurocognición y Psicofisiología (NeuroLab), Universidad Iberoamericana, Santo Domingo, Dominican Republic, **3** Department of Community Health, College of Medicine, Federal University of Ceará, Fortaleza, Brazil, **4** Department of Social, Behavioral and Population Sciences at the Celia Scott Weatherhead School of Public Health and Tropical Medicine at Tulane University, New Orleans, Louisiana, United States of America, **5** Department of Psychiatry and Pediatrics, Boston University Chobanian and Avedisian School of Medicine, Boston, Massachusetts, United States of America, **6** Department of Psychiatry, Boston Medical Center, Boston, Massachusetts, United States of America, **7** School of Nursing, University of Texas Medical Branch, Galveston, Texas, United States of America, **8** Department of International Health and Sustainable Development, Celia Scott Weatherhead School of Public Health and Tropical Medicine at Tulane University, New Orleans, Louisiana, United States of America

\* adrianne.nelson@yale.edu

## Abstract

### Background

In the Dominican Republic about 14.5% of children do not reach their full potential by age five, with children of low socioeconomic position most affected. The Nurturing Care Framework is an evidence-informed actionable framework to help children thrive, but we must first understand cultural contexts and childrearing practices that contribute to delay. This study applies the Nurturing Care Framework to explore the context of home-based care among young children in the Dominican Republic.

### Methods

We conducted a sociodemographic survey and semi-structured qualitative interview with 25 mothers ages 19–42 (7 under the age of 18 at first birth) with low socioeconomic position and children under five that live in the capital city Santo Domingo. We asked in-depth questions about the Nurturing Care Framework's domains of responsive caregiving and opportunities for early learning. We used consensual coding and deductive thematic analysis to analyze transcriptions, examined convergence and divergence in themes between adolescent and adult mothers, and organized themes using concept mapping.

**Data Availability Statement:** Data cannot be shared publicly because they are identifiable qualitative data and cannot be adequately de-

identified. Additionally, following the ethics committee protocol approved by Tulane University (2019-2375), the audio files and transcripts, which contain sensitive information and are provided in conditions of confidentiality, cannot be shared. This follows the recommendations made by Martijn de Koning and colleagues in "Guidelines for anthropological research: Data management, ethics, and integrity" (Ethnography, 2019. 20(2): p. 170-174). Data are available upon request to the Tulane University Institutional Review Board for researchers who meet the criteria for access to confidential data and with a complete Data Use Agreement. Approval can be requested from, and data provided by, Roxanne Johnson, Director of the Human Research Protection Office at: rjohnson@tulane.edu. Data requests can also be sent to the corresponding author and are contingent on approval from the Director of the Human Research Protection Office.

**Funding:** This work was supported by a grant from the American Fulbright Association. The funder had no role in the study design, collection analysis, or interpretation of data, writing of the report, or decision to submit the article for publication.

**Competing interests:** The authors have declared that no competing interests exist.

## Results

A few mothers provide responsive caregiving to their child, but they are unaware of its benefit to their child's development. Adolescent mothers expressed lower confidence in their mothering skills. Across age groups, mothers did not see themselves as agents of change in their child's early learning process and allow several hours of videos each day. Mothers provide children opportunities for learning through social interaction, a possible strength among this population. With regards to security and safety, about half of mothers use corporal punishment, all but one of these is an adolescent mother.

## Conclusion

Findings highlight the need for parenting programs that build on strengths such as child-to-child social interaction and provide parents with opportunities to develop knowledge and skills to provide early learning opportunities. Interventions should target families with low socioeconomic position and adolescent mothers.

## 1. Introduction

In 2016, the Lancet published an article showing that almost 250 million children in lower- and middle-income countries do not reach their full potential due to poverty-related factors [1]. The impact of poverty on the child's brain is cumulative, wherein increased deprivation is associated with deeper developmental delays [2,3]. Poverty and other associated factors impact early brain development through several pathways known collectively as adversity [4]. This includes inadequate nutrition and disease, which can lead to stunting and wasting [5]. Abuse and neglect are also associated with lasting effects on a child's development and emotional well-being [6]. The consequences of adversity can reach adulthood, impacting the ability of a person to have healthy relationships and hold jobs [4].

### 1.1 The Nurturing Care Framework

The Nurturing Care Framework (NCF), developed by the World Health Organization (WHO) and its partners in 2018, sums up decades of early child development (ECD) research in a comprehensive approach to promoting children's optimal development and well-being [7]. The NCF focuses on five necessary components of early care: 1) adequate nutrition, 2) good health, 3) security and safety, 4) responsive caregiving, and 5) opportunities for early learning. Adequate nutrition involves vitamin supplementation during pregnancy, exclusive breastfeeding to at least age six months when possible, and a diverse diet free of toxins and parasites. Good health involves a caregiver protecting a child's physical and emotional health, protecting a child from environmental dangers, having good hygiene practices, and providing appropriate medical care when needed. Security and safety involve helping young children feel safe, even when they misbehave, and protecting them from physical danger and emotional distress by preventing accidents such as choking or falling and avoiding physical punishment. The caregivers' responsibilities involve an awareness that a young child can experience intense fear when beaten or threatened with abandonment, and that this fear can interfere with a child's ability to develop trusting relationships in the future.

The NCF's emphasis on responsive caregiving distinguishes it from other early childhood models. Responsive caregiving, defined in the model as "observing and responding to

children's movements, sounds and gestures, and verbal requests", begins early and provides the foundation for neurodevelopment and learning. Responsive care is a buffer for children experiencing early adversity [8,9]. This form of caregiving is particularly relevant in settings where children are impacted by adversity in daily life: A as a growing body of longitudinal research demonstrates that responsive caregiving plays a role in mediating the effect of poverty on child development [10]. For example, in 2019, Luby et al followed preschoolers for 5–10 years to assess how poverty in pre-kindergarten impacted brain development in elementary school. They found that poverty was associated with smaller white and cortical gray matter and hippocampal and amygdala volumes, the areas of the brain collectively responsible for translating emotion into long-term memory. Parental caregiving support, measured observationally during a preschool visit, mediated this relationship [10].

Closely related to responsive caregiving is early learning. Infants need visual, auditory, and tactile stimulation through light, sounds, visual patterns, objects, and tactile experiences in the environment [11,12]. Most of this learning will occur in the context of consistent and affectionate interactions with adult caregivers. Together with adults, infants will explore their surroundings, seek out visual and tactile stimulation, share attention on objects with others, orient towards and listen to voices and singing. As they grow older, they may also participate in preschool or social interaction with other young children in stimulating environments [13]. Given their importance, researchers have integrated early learning and responsive caregiving into parent and caregiver/educator coaching interventions in settings of poverty and seen positive ECD outcomes [14].

The NCF employs a life course perspective within which the critical period of development is from pregnancy to age five [13,15]. This is based on research showing that, during this time, a 'window of opportunity' allows the brain to construct vital internal structures. The brain develops thousands of neural connections, and its plasticity allows children to absorb vast amounts of new information and experience rapid brain growth [16]. Around age five, weaker neural connections are discarded, and stronger ones are strengthened to allow for more efficient brain function in a process called 'pruning.' During the window of rapid growth, it is crucial for children to develop basic brain architecture and processes that they will then strengthen, expand, and build upon in primary school and into adulthood [16].

Further work to understand responsive caregiving and opportunities for early learning in the DR, therefore, could help ease the impact of poverty on children's ability to learn and thrive. This study aims to develop knowledge about how mothers living in poverty in the DR provide their children with nurturing care, according to its conceptualization within the Nurturing Care Framework. In particular, we explore caregivers' beliefs and practices regarding responsive caregiving, opportunities for early learning, and safety and security in the home.

## 1.2 Culture in ECD research

In 2022, The Lancet Child and Adolescent Health published a response to a paper by McCoy et al claiming almost 75% of children in lower- and middle-income countries were not receiving minimal nurturing care [17]. The response, titled "Different is Not Deficient: Respecting Diversity in Child Development," called attention to the importance of appreciating difference in cultural childrearing practices [18]. It points out that NCF indicators used in large-scale assessments may be missing facilitators to early development in low- and middle-income countries and thereby inadvertently stigmatizing these families. For instance, McCoy et al determined that early learning is present if the family owns a book or toy, and the child is attending an early child education program [17]. These indicators don't include other early learning opportunities, such as playing with older children, singing, playing with everyday

household items or recycled toys, and reading non-print material such as signs. McCoy et al point out that the NCF is a useful framework and should be interpreted within each cultural setting with care. Likewise, ECD assessment tools fall short of defining domains that reflect daily life in settings with high concentrations of poverty such as the Dominican Republic [19]. Understanding these domains using culture-specific data will allow for feasible action steps toward achieving goals such as ensuring lifelong learning and eradicating poverty set by the United Nations as part of the 2030 Sustainable Development Goals [20].

### 1.3 ECD in the Dominican Republic

The DR was chosen for this study due to the high prevalence of poverty and other socioeconomic hardships impacting neurodevelopment in children. The Dominican Republic (DR), a Spanish-speaking Caribbean nation on the island of Hispaniola, has a population of 11.2 million people with 49.3% of children living in poverty, according to a 2019 government report [21]. In the DR, the prevalence of neurodevelopmental delay, defined as not reaching full developmental potential by age five, is high at about 14.5% [22], compared to 4.6% in the United States [23]. This reality is associated with school achievement: based on the 2014 and 2019 Multiple Indicator Cluster Studies (MICS) data, primary students in the DR perform among the lowest in Latin America on math and language evaluations [19]. Further, in 2022, Sánchez-Vincitore and Castro found that socioeconomic gradient predicted ECD outcomes among children 36–59 months of age and that a psychosocial model, including opportunities for early learning components, mediated this relationship [24].

Since 2014, the government provides low-income children with early stimulation, home visits, and preschool, and women with high-risk pregnancies accompaniment to local hospital visits, identification registration, and government-sponsored social programs providing a minimum monthly salary, vaccinations, and food baskets through the *Programa Integral a la Primera Infancia de Base Familiar y Comunitaria* [25]. The *Instituto Nacional de Atención Integral a la Primera Infancia* (INAIPI) provides preschool services, including preschool for children ages 45 days to five years, similar to the Head Start program in the United States. In 2021, INAIPI reached 215,933 children ages zero to five years [26].

## 2. Methods

This is a qualitative study using thematic analysis to interpret patterns and identify meaning within the data. We chose thematic analysis because it helps to understand people's experiences, views and opinions. It is useful for comparing and contrasting patterns between groups, such as in our case, between adolescent and adult mothers [27].

### 2.1 Recruitment

Eligibility criteria included having a child under five years of age, reporting an income less than $3 a day, and living in and around Santo Domingo, the capital of the DR. We recruited participants from a contact list of women who had participated in the Pregnant Women-Centered Care in the Dominican Republic Project (Project MAC). This was a Tulane study funded by UNICEF that identified obstetric violence during childbirth through birth observation in two public hospitals in Santo Domingo and a neighboring province [28]. All participants had agreed to be contacted by phone for future studies. For the UNICEF study, birth observers were present for all births during a two-month period in October and November 2019 without any other exclusion criteria. Approximately three years later, the first author, together with two research assistants, sorted the 169 addresses of women who lived in Santo Domingo by distance from the Laboratorio Laboratorio de Neurocognición y Psicofisiología (NeuroLab) at

the Universidad Iberoamericana, where the team was stationed. The research team called phone numbers beginning with those who lived closest and working out. Approximately 112 phone numbers were out of service or no longer belonged to the previous owner, 36 did not respond to contact attempts (one text and two phone calls for each number), seven were initially interested but did not respond to further requests to set a date, one said she did not have time for the study, and two had lost their child since birth and had no other young children. Thirteen women from the previous study agreed to participate in interviews, all from the province of Santo Domingo. To augment our interviews and provide contrast, we used snowball sampling to recruit an additional 12 women from a town called Yamasá, located about an hour from Santo Domingo in the province of Monte Plata. Ultimately, we conducted interviews with 25 women who met eligibility criteria, 10 women were from Los Mina in the province of Santo Domingo, three from Boca Chica in Santo Domingo, and 12 from Yamasá.

## 2.2 Participant consent

All participants consented to take part in the study and authorized the utilization of their de-identified data for research purposes. Those who joined the study via Zoom provided verbal consent, which was captured and recorded during the interview. The researcher obtained consent signatures from these participants by asking them to permit and witness an electronic rendering of their name on the consent form. We recorded interviews using the voice recorder application on the researcher's laptop or phone. One participant preferred not to have their interview recorded, and the interviewer took notes instead. The two research assistants transcribed the interviews verbatim in Spanish, removing all identifying information and designating a pseudonym for confidentiality.

## 2.3 Data collection

Because most children under five in the DR are at home with a female caregiver during the day [21], we focused the interviews on the home environment. We asked about responsive caregiving and opportunities for early learning in-depth and intentionally allowed conversations about security and safety to emerge organically during the interviews. We chose to target these topics and not good health and adequate nutrition because the former have yet to be explored in the literature among this population and evidence suggests these areas may buffer the impact of poverty on child development [8,9]. In contrast, national-level data are available about the health and nutrition of this population [21]. Finally, responsive caregiving and opportunities for early learning are well-suited for integration as parent coaching modules in existing public health interventions and of interest among our partners at INAIPI.

## 2.4 Interview guide

We used semi-structured interview guides, with open-ended questions taken from the Working Model of the Child Interview and informed by the Nurturing Care Framework. The Working Model of the Child Interview is a structured interview designed to identify a mother's internal representations, or working models, of the relationship with their child [29] (S1 File). We used nine of ten questions from the WMC (numbers 1 (simplified), 2, 3, 4, 5 (simplified), 6 (simplified), 7, 9, and ten). We simplified several questions and removed questions soliciting specific words. We then added in additional questions related to responsive care and opportunities for early learning. For each of these domains, we inquired about 1) how mothers interact with their child when they are alone, 2) their perception of what experiences and behaviors promote development, and 3) how they perceive the child's environment and interactions to impact their child's development. Before the interview, we administered a short structured

sociodemographic survey to gather information about number and ages of children. We also included one question from the Household Food Insecurity Assess Scale that asked how often they had worried about not being able to afford food for their family in the past month, with three Likert scale response options (1- Rarely or 1–2 times in the past month, 2- A few times, or 3–10 times in the past month, and 3- Many times, or more than 10 times in the past month) [30]. We conducted data collection between August 13, 2022 and February 15, 2023.

The first author (principal investigator) trained two Dominican psychologists research assistants to conduct qualitative interviews. Research assistants observed five interviews conducted by the first author, then the research assistants conducted three interviews with the first author present. Research assistants conducted a total of seven interviews. We ensured all interviewees could speak privately during the interview. We conducted all interviews in Spanish, in-person (N = 21) and virtually (four women had moved to distant locations after participating in the original study). We conducted all interviews in quiet, private locations. In Santo Domingo, we held in-person interviews at public meeting places selected by participants (e.g., restaurants, shopping malls). In Yamasá, we conducted interviews in the patio or yard of participants' homes. The participants received a stipend in consideration for their time.

## 2.5 Data analysis

We used deductive thematic analysis to analyze transcriptions based on themes from the Nurturing Care Framework and the Working Model of the Child, but allowed for additional themes to emerge from the data [27]. Literature indicates that parent age is associated with parenting behavior [31,32], so we examined divergence and converge among data from adolescent versus older mothers. First, the first author and the research assistants conducted open coding with a pre-developed codebook in Dedoose [33]. We structured the codebook to cover the Nurturing Care Framework's themes of responsive care and early opportunities for learning and allowed subthemes to emerge from the interviews. When it was unclear which theme an excerpt fit in, we consulted the Nurturing Care Framework's summary. For example, this quote involves both responsive caregiving and early opportunities for learning: "*sometimes I help her, I sing. And I teach her the vowels, I give her a pencil and I say, 'look, write here, this is the letter A. Look, these are the numbers, count them'. She starts to count.*" We coded this excerpt as responsive caregiving because the action subject is the mother and her interaction with the child. The child's response is secondary. Because participants simultaneously reported on safety and security topics, we included that as a theme in the codebook. Our final themes included: 1) responsive caregiving (subthemes: relevance for development, parent interactive behavior, perceived role as mother), 2) opportunities for early learning (subthemes: beliefs about how children learn, intentional teaching, preschool/ early stimulation), and 3) safety and security (subthemes: managing difficult behavior, use of corporal punishment). All analysts individually coded five interviews while blind to others' coding and then compared answers. After five interviews the three analysts reached 80% agreement and we proceeded to code individually, checking our coding every five interviews to avoid drift. Once all interviews were coded, we wrote summaries of each theme, and organized the themes into a logical narrative [34]. The first author and one of the assistants separately translated Spanish excerpts into English for this report and discrepancies in translation were resolved by the senior author, a native Spanish speaker with 10 years' experience conducting maternal and child health research in the Dominican Republic.

## 2.6 Ethics review

We received approval from the Universidad Iberoamericana Human Research Protection Program (# CEI2022-13) and the Tulane University Research Ethics Committee (# 2019–2375).

## 3. Results

### 3.1 Demographics

Participants' mean current age was 27 years "Table 1". Seven participants became mothers before reaching the age of 18 and mean age of motherhood was 19.5 years (median = 18 years). They had a mean of two children and the youngest child was two years old. Three participants had children that did not live with them, in addition to those who were living with them, and two women were caring for dependents or children in their home who were not their biological children. Eleven had at least a high school degree, 16 were employed, and three women stayed home exclusively with children. Eighteen participants reported food insecurity, defined as being worried about having enough food for their family in the past four weeks.

### 3.2 Results summary

With respect to the relevance of responsive caregiving for development, few women explicitly identified mother/ child interaction or responsive caregiving as important for their child's development. However, many spontaneously described taking part in this behavior. With regards to their role as mothers, many expressed intentions to raise well-behaved children. Some participants, particularly adolescents, demonstrated low confidence in their mothering skills. Speaking about opportunities for early learning, participants mostly expressed confidence in tools external to what they could provide themselves, such as preschool or educational videos. A few mothers took part in intentional teaching, often together with educational cartoons or games on the tablet with material related to colors or numbers. Mothers voiced confidence in their child learning from interactions with children of all ages, such as in a preschool or informal family setting. When addressing issues of security and safety, participants reported struggling with their child's behavior and almost half of mothers reported using corporal punishment, a phenomenon more common among adolescent mothers. See "Table 2" for additional themes and selec´´´ted quotes not presented in the text.

**Table 1. Demographic data, n = 25.**

| | N or mean | % or range |
|---|---|---|
| **Current age** | 27 | |
| **Age at first birth** | 19.5 | 15–31 |
| **Under 20 at first birth** | 17 | 68% |
| **Under 18 at first birth** | 9 | 36% |
| **Married / living together** | 11 | 44% |
| **Number of children** | 2 | 1–5 |
| **Education** | | |
| Grades 1–8 completed | 2 | 8% |
| Elementary school completed | 1 | 4% |
| Some high school | 4 | 16% |
| High School completed | 11 | 44% |
| Some university | 6 | 24% |
| University completed | 1 | 4% |
| **Employed** | 16 | 64% |
| **Reported food insecurity in the last month** | 18 | 72% |
| 1–2 times | 6 | 24% |
| 3–10 times | 5 | 20% |
| >10 times | 7 | 28% |
| **Reported using physical punishment with their child** | 11 | 44% |

**Table 2. Select excerpts.**

| Theme | Quote | Quote |
|---|---|---|
| **Responsive caregiving** | | |
| Relevance for development: how important caregivers believe responsive caregiving is for the healthy development of their child or others of similar age. | *Interviewer: What are the best ways for a girl that age to develop? What does she need, in your opinion? Participant #2: The best way? That her mother interacts with her. I say: "I'm going to play with the girl." It's not that she's going to jump and run, not here. But she can grab a ball and say, "throw it, mom." And the girl throws it: "One, Two. . ." or something. If I can, I'll teach her the numbers through that game. and so on. Interact with the child. . .there are many children who are slow to develop. And it's not because of their mentality, it's because they leave it like that, make them learn alone. My boy, since he was little, I teach him, I explain to him, all the time.* | *Participant #11: She and I. . . she likes us to play together. Sometimes she puts makeup on me and brushes my hair, I brush her hair. We play a lot. I want her to trust me, I hope she trusts me a lot, and she tells me everything that is going on in her life. Whenever she wants to play and she has nobody to play with, I play with her and we pass the time playing together. We play together a lot. Interviewer: When you play, you would like to build trust with her, so she tells you things. Participant #11: Yes, so that she considers me a friend.* |
| Parent interactive behavior: opinions and examples of interactive behavior with child. | *Interviewer: When you play, for example, singing or playing doctor, do you play too? Participant #4: Of course, I play with her. I bought her a doctor set and we play. She says to me, "Mama, let's go to the doctor" and I say, "we are taking you to the doctor so they can heal you". Then she says, "No, not me, we are taking you to the doctor". She says, "mama, lay down there", and I lie down, and she puts a bandage on me to heal me. She has her bandages and heals me with them. I say, "you are the silliest doctor, you put a bandage on everything". And she says, "yes, mama, let's heal you quickly". My girl, she is a good companion, she can't stand to see me in pain, she can't watch me cry either.* | *Interviewer: Do you think there is anything you can do while interacting with her at home to help her grow and learn? You mentioned that she learns things at school, do you think you can do anything in the home? Participant #5: Yes, sometimes I help her, I sing. And I teach her the vowels, I give her a pencil and I say, "look, write here, this is the letter A. Look, these are the numbers, count them". She starts to count. She counts a lot and sometimes she watches cartoons that have numbers. They say, "look, here is 1", and she says, "one", and then I say, "two".* |
| Perceived role as mother: the mother's parenting approach, dreams and values, and how they transmit them to their child during interactions. | *Interviewer: What are the dreams that you have for your daughter? Participant 14: My dreams? There are many, there are many. I dream of being somebody else so I can give her a better life. That when she is my age, that she can go to the university without any problems or worries. But I have no money. I have no job to continue studying. I mean, that's why I'm trying to make an effort to move forward.* | *Participant #17: There is a cartoon, it is called Masha and the Bear, and she is very spoiled. When she started watching it, [my daughter] began acting out, I had to take away the tablet, the telephone. I had to tell her father not to give her the phone either, he would just put whatever she wanted on the phone. Interviewer: Right, because Masha acts out sometimes. Participant #17: Exactly, she is very, very bad. [My daughter] began a bad, terrible phase. She hit me a lot, very bad.* |
| **Opportunities for early learning** | | |
| Beliefs about how children learn: the mother's opinion on how she can best encourage learning and development for her child. | *Interviewer: Is there anything that you do to help him grow and learn? Participant #3: Well, I put cartoons on the phone, educational cartoons, and he repeats them and sings. I always repeat them too, "do this, say that", and he repeats it.* | *Participant #2: I teach her songs. La cucaracha, I don't know if you've heard it, she sings it in its entirety. The cartoons she watches are not from here, they are foreign. They are for child development. Participant #2: The dolls say something, and I repeat it to them. For example, "What color is that?" and she answers me: An eye", no, [I say], Purple" and those cartoons have helped her a lot. But her development is normal for her age. I mean, it is very advanced.* |
| Intentional teaching: examples of a mother's purposeful teaching to encourage her child's development. | *Participant #17: I try to teach her, for example, numbers. She already knows how to count to 10, she knows the vowels, she already knows all the colors and I try to show her educational cartoons to open her mind so she learns more. She is going to be a very open girl; you can't talk much in front of her because she says everything.* | *Interviewer: Well, I have one last question. What do you think are the best ways to help young children develop? Participant #20: Well, be very patient, because little children move a lot. Always keep an eye on them, make sure they are clean. Give them their food, their breakfast. Mine likes that game so much, he is always asking for juice, juice, juice. Always be attentive to them, when children move a lot, you have to be attentive to them so that they do not go to pick things up from the street, so that they do not crash into them. Always be attentive, watching what they do.* |

*(Continued)*

**Table 2.** (Continued)

| Theme | Quote | Quote |
|---|---|---|
| Preschool / early stimulation: any involvement in developmental activities outside of the home. | *Interviewer: Does the girl go to daycare, school? Participant #17: No, she is not going to school right now because I am trying to teach her to read and write at home.* | *Interviewer: Ok, you don't let someone else take care of him. Participant #6: No, never. His grandmother, nobody else. Well, now I'm going to start working on Friday. I'm thinking about taking him to a daycare facility. I'm getting advice on what it's like, what's done, and the environment, to see if I can leave him there. It's really a little more difficult for me to move near work rather than to leave him there. For now, I'm going to take him for a half day, from 8:00 A.M. to 12:00 P.M. and from 12:00 P.M. to 4:00 P.M he is going to be taken care of [by his grandmother], so that he doesn't get too stressed.* |
| **Safety and security** | | |
| Managing difficult behavior: the mother's approach to dealing with a child when they misbehave, including punishment. | *Participant #18: The truth is, she is half bipolar because she has a day when she behaves extremely well, a day when she behaves badly, so badly that I don't know how to control her and I wouldn't like to. . . Sometimes she behaves so, so bad, I mean. I don't like hitting her, but I have to breathe and sometimes count to a million because it's not easy. She has a rather abrupt personality, if she says she wants something, you have to give it to her. I don't want to raise her like that, people will say that I don't educate her and I'm ashamed.* | *Interviewer: When you're upset, what do you do? Participant #19: Sometimes yes, sometimes he gets spoiled when he wants something, and I don't want to give it to him. Interviewer: What do you do when he gets like that? Participant #19: I punish him, grab him, and make him kneel. He has to stay like that for 5 min. Interviewer: Do you hit him too? Participant #19: No, no, I don't hit him. He is very small; I don't even hit older children. Interviewer: Where did you learn this thing about kneeling? Participant #19: From my dad and mom. When [my son] does something bad, I tell him: "Go and kneel!" and he grabs his knees and kneels. Sometimes I even tell him: "It's ok, don't worry!" but he grabs his knees and kneels anyway. . . If you let the child do whatever he wants, when he grows up it will be a problem for you. He will say, "when I was little, my mother let me do whatever I wanted, I'm going to do whatever I want now!" Limits must be set for children. If you don't set limits, then it's a headache for you later.* |
| Use of corporal punishment: a mother's report of using corporal punishment with her child, including her opinions about its use. | *Participant #8: The crybaby, that's what makes me very angry *Laughs*. I can't hear a child crying much. I try to control myself. Before, I couldn't control myself, but with the passage of time I have learned how to control myself and I cope with it. Before, I really couldn't cope when he was a baby, I used to push him, I pushed him. I know that was bad, it was bad, because he cried about everything. Since it was my first time, I didn't know how to handle it. I'm not a child's person, you know?* | *Participant #2: Sometimes [my daughter] is in bed and wants milk or water. Sometimes I am in the middle of giving her food and she demands water. Sometimes I can't stop at that very moment, and she goes and breaks all my bottles. She breaks the bottles by hitting the table. She got into trouble recently because she smashed the bottles and hit the television screen. It made me go crazy. I hit her. When she crashes and breaks things like that, I give it to her.* |

## 3.3 Responsive caregiving

**3.3.1 Relevance for development.** Participants did not spontaneously relate responsive caregiving to early child development. Only two mothers mentioned responsiveness. This mother talked about being in tune with the child's mood and emotions and following up with her.

*Participant #21: I tried to understand them, they understand what you say, at least I believe so. Ask them what they have in their hand. The other day I realized another child was pulling my son's hair. Pay attention, dedicate time, that is important. One notices, I take my daughter [to school] and I pick her up. When I'm not there, my cousin or my father, or sometimes my*

*mother when she is free, picks her up. If my cousin says she is acting weird, and gets home and doesn't talk, something is off.*

**3.3.2 Parent interactive behavior.** Participants rarely mentioned responsive caregiving as essential for child development. However, several reported practicing intentional interactive conversation and a few, such as this one, said they did so to promote their child's development:

*Interviewer: What do you think is the best way for a child his age to develop?*

Participant #19: How so?

Interviewer: What are the activities that an adult could do to promote a child's development and abilities?

*Participant #19: Interact, the more you talk to a child, the more intelligent they will be. If you don't talk with your child from the beginning, he won't talk or know. I talked to my children from pregnancy, beginning when they were in my belly. Their grandmother also, their paternal grandmother. Sometimes he would move and such, and she would say, "what's going on with the little boy? Why is he moving around and giving mommy a hard time?"*

While most women reported communicating with their children using interactive dialogue and play, most viewed these interactions as a way to show affection, rather than a learning tool:

*Interviewer: When you are alone, just the two of you, do you play? Do you play with her sometimes, or does she mostly play alone or with other kids?*

*Participant #5: No, we play sometimes, and we talk. I lie in bed with her, "tell me, do you love me? Do you love mommy?" "Yes," and we talk and talk in bed sometimes until we fall asleep.*

*Participant #7: Yes, there are times when I sit on the floor when he is distracted with something else and I say, "let's play with the ball" and he sits, opens his legs for me to throw it to him, and he throws it back to me. We play that way for a while.*

Given many women did not associate interaction with play, it may be understandable that when asked about play, one participant responded about providing her child with a nutritious diet. This may imply that, regarding the facilitation of child play, some mothers conceptualize their role as ensuring the child has adequate nutrition and energy to play rather than an active participant in and instigator of play.

**3.3.3 Perceived role as mother.** To understand participants' goals and values related to child raising, we asked mothers about their dreams for their child, and how they guide their children toward achieving them. Mothers had difficulty articulating a parenting approach, future dreams, or expectations for their child. Most mothers mentioned values such as staying out of trouble, knowing the difference between right and wrong, and being kind and loving. Almost all mentioned not wanting their child to be spoiled.

*Interviewer: When you think about [the child's] future, do you have any dreams for her life?*

*Participant #5: Well, I would like her, all three of [my children], to be good, obedient, and listen. And that they don't go down the wrong path like other children. . . . As a mother, I would like her to make her own decisions, the right ones, go to church, study. That's what she wants*

*to do in the morning. I hope she's not like those girls wandering the streets. A good girl. That is what I hope for as a mother.*

Some adolescent mothers expressed little confidence in their skills as mothers, mentioning that other family members were better caregivers to their child than themselves:

*Interviewer: Ok. When he is not at school, you said that your mother takes care of him. How is she with your children*?

Participant #16: My mother takes better care of them than I do.

*Interviewer*: *How so*?

*Participant #16*: *Because she sometimes buys him things, she buys him yogurt, this and that, and sometimes, you know, I don't have much money to be buying things. My mom attends to all of my kids' cravings. If they don't want rice, she goes straight to the store to get something else, that kind of thing.*

In cases where children were fostered to other family members, mothers said it was because she worked full time, her house was far from school, she did not live in a safe neighborhood, or she trusted them more to care for her children.

## 3.4 Opportunities for learning

**3.4.1 Beliefs about how children learn.**   When asked how they could help their children learn and grow, women talked about providing their children with economic opportunities (e.g. paying for school supplies, transportation, and accommodation) and teaching them right from wrong. When asked how they knew their child was developing correctly, mothers emphasized crawling early, drinking from a glass, and speaking clearly and with confidence:

*Participant #3: At four months, he was crawling on the bed. Just like that, and I said, "oh, he crawled so early, and stopped breastfeeding," he didn't need that either.*

Interviewer: Wow, he is so independent.

*Participant #3*: *Yes, and I said,* "*you think you're a man, don't you*! *Drinking from a glass*!*"*

Interviewer: He is advanced, does he have a lot of words?

*Participant #3*: *Yes, he spoke early too.*

**3.4.2 Intentional teaching.**   Many participants described preparing activities for the child for entertainment purposes. For instance, they use electronic devices to watch videos, usually cartoons on TV, tablet, or a parent's cellphone. Mostly, it is for extended time unsupervised:

*Interviewer: When you two are together, do you play with him?*

Participant #10: Yes.

Interviewer: Or does he play by himself?

Participant #10: No, we always play lying down, playing and I always play with him. I watch Tik Tok in the tablet with him.

Interviewer: Does he initiate the play time?

Participant #10: Yes.

Interviewer: Or you bring something for you to share?

Participant #10: No, he does, I do, we both always do.

Interviewer: Both of you initiate the game. And you play for a while. Is there something, in particular, you look to achieve when you play with him? Are you thinking you'd like him to learn anything in particular or just pass the time?

*Participant #10: No, because he is always playing on his tablet in the house, watching cartoons, stuff like that.*

However, many mentioned using the tablet for educational games that teach vowels, numbers, body parts, colors, and fruits. A few mothers described taking part in intentional teaching with their child while on their tablet. Only two participants mentioned reading books together. One mother interacts with the child while the child plays educational games:

*Participant #4: She has her tablet, which has many educational games also. It teaches her colors and names of fruit. It teaches her many things. I say to her, "what color is this?" and she says, "yellow!" and I say, "good job!" I come back and repeat it, and when she does the tests, I help her, and we review together. It's not just having her do it, I show her how and she does it, so that she can learn.*

The few mothers who mentioned taking part in intentional teaching with their child focused on specific topics. For instance, one teaches her child about personal hygiene, another Catholic beliefs, and another uses toys to animate teaching. Almost all caregivers reported having at least one toy in the house for the child:

*Participant #18: She gets up, and she always asks for something, always. I give her juice, I wait a few hours, then I bathe her, and she eats breakfast. She starts to play, and sometimes she likes to play with colors. A song says, "Mommy finger, mommy finger, where are you?", and I reply, "Here I am, here I am". And then she asks, "what color is this?" and I tell her the color. We play with dolls, she has blocks, and we play with blocks. I play with her, but as her father plays, nobody else in the world does.*

Interviewer: How do they play?

*Participant #18: He gets home, he plays with blocks, colors, he brings her special crayons for children that don't hurt her hands. He doesn't mind, they play for so long. When he gets home, they start to play and they play together until she falls asleep, until she is exhausted from playing.*

Several women indicated their child likes to play with blocks, balls, and cars by themselves:

*Interviewer: Do you play with your daughter?*

Participant #21: Yes, she has her games, like the piano. She really likes blocks, to paint, as long as it is not on the walls. She barely plays with dolls at all, she likes crafts, stuff like that. She really likes to paint, blocks, and her piano.

Interviewer: Do you play those games with her?

*Participant #21: Yes, the blocks and the piano.*

Interviewer: Who initiates the play?

*Participant #21*: *Usually her, I am not going to lie, I get home tired. I dedicate a little time, but I have to cook, etc. I always try to give her a little time.*

**3.4.3 Preschool/Early stimulation.** About half of the children do not attend preschool. These children are mostly cared for by their mother or another family member, most commonly a grandmother. Of those that attend preschool, a few attend a half-day preschool, a few attend an INAIPI center—the government-subsidized early child stimulation program—, and several attend a full-day school. Just one child attends a *casa hogar*, or a daycare in a private home. One child was not accepted into preschool because his government-issued ID was assigned to someone else, and the mother was working on completing the paperwork to resolve the issue. Most mothers mentioned they believed that their child was too young to attend preschool.

Overwhelmingly, children played with other children, and mothers expressed their belief that playing with other children helped children learn and develop. One mentioned that her child plays well with other siblings because he is smart for his age:

Interviewer: What do you believe is the best way for a child to develop?

*Participant #3*: *They develop quickly when they are in an environment with another child. They learn faster, speak clearer, and do things they don't know how to do. With others, not when they are alone.*

About a third of participants mentioned that the child plays with cousins, neighborhood kids, and friends at school. A few wished their children had more opportunities to play with other children but didn't because they live in a neighborhood that is unsafe or far from relatives.

## 3.5 Safety and security

**3.5.1 Managing difficult behavior.** Caregivers overwhelmingly focused answers to questions of all sorts on discipline and the use of corporal punishment. About half of the caregivers mentioned the child's behavior being difficult to manage, and a few mentioned that leaving the child with another caregiver was a challenge because the child did not want to separate from them:

*Interviewer: Do you enjoy your relationship?*

Participant #5: Yes, we enjoy it, because I am with her, sometimes we play together. We are always together because she doesn't leave my side, ever. Sometimes I tell her, "Come, you are going to stay with your grandmother," at her grandmother's house, and she doesn't like to, she doesn't like to.

Interviewer: Why do you think she is like that?

*Participant #5*: *It must be because she is never far from me. All the time since she was born near me, she is always by my side. She has never left me. If I go somewhere, she goes too. Always by my side.*

Two mothers mentioned that they were worried about their children's aggressive behavior:

*Interviewer: How is he different from other children?*

*Participant #13: He plays, but he doesn't play like other kids, like the kids who pick up a toy car and play with it for the entire day. He doesn't play, he likes to bully other kids. When he and his friend get together, they fight. He isn't having a childhood full of toys and games, not him. He just looks for fights.*

One mother mentioned that she considers her child to be "bipolar" and with a strong temper. Another participant said that her child, age three, does not enjoy physical touch (hugs, kisses, cuddling). She mentioned that she respects his boundaries but complained that the child behaves like his father:

*Participant #9: He is like his father; he doesn't show his feelings. He doesn't like to be hugged or kissed. He doesn't even like me to dress him, he doesn't like any of that. I don't know where this behavior comes from. . . his father is like that too.*

Interviewer: How do you feel when he does that?

Participant #9: I feel strange, because the other two older kids, they just want me to hold them, kiss them, snuggle them. But not him, he doesn't like it, he doesn't have that kind of love.

Interviewer: Right, he doesn't express himself that way.

*Participant #9: He doesn't know how to express love, just like his father.*

Another participant explained that she was exhausted by the child's tantrums regarding school and personal hygiene. A mother also described the child as hyperactive due to sugar overconsumption:

*Participant #17: One day we were in the pharmacy, and she was having a fit because she wanted a cake, and I don't like to give her too many sweets because she gets intense when I give her sweets. I told her no, that I wasn't going to buy it for her, and she started kicking and hitting and saying yes, that she wanted me to buy her a sweet. And I told her the police would come to get her and I told the security guard, "police, come get her" and the guy, playing with her, pretended to take her and she started screaming and screaming, and until today she hasn't wanted anything to do with the police at all.*

**3.5.2 Use of corporal punishment.** With respect to discipline, almost half of the women (11) mentioned *dar pela*, or hitting her child when they misbehaved, but felt regret after doing so. This was more common among adolescent mothers where 10 of the 17 who were under the age of 20 at first birth spontaneously mentioned using physical punishment, compared to one of the eight of the women who first gave birth at the age of 20 or later.

Mothers described feeling impatient with the child when they act out and having difficulty controlling their urges to use physical punishment excessively. A few women believe that *dar pela* helps build character and not be spoiled. Several mentioned receiving physical punishment as a child and trying to control the impulses now, but not always succeeding:

*Participant #9: It affected me because I don't want to be like that, but I am sometimes. I can be aggressive, and I can take things the wrong way. I try to control myself; I try not to be [like my mother] because I don't want to be like her, but sometimes I take things the wrong way, and I get aggressive, I want to fight, I want to scream, but I control myself. Sometimes I can't control myself. Sometimes I can be very aggressive, sometimes a bit aggressive with the*

*children too, that's why I sent them to live with my parents. The same things my mother did to me, I was starting to do with them. Biting their hands so they would be quiet; when I wanted them to be quiet, I would bite their hands. I thought, "Please God, forgive me." It wasn't what I wanted; I don't know why I did it.*

One mother said she hits her child only when she believes her daughter deserves it. Two others threatened to hit their child but didn't mention actually doing it. One mentioned she yells but does not hit. Of those who specifically mentioned not hitting their child, one tells herself the child's bad behavior is because they are not having their needs met—they are hungry, tired. Another believes that talking is better because hitting can generate more bad behavior. Several others mentioned taking away privileges such as the tablet, putting them in timeout, and simply refusing until they tire:

*Participant #6: You know, they are like animals. For example, animals don't use language, and what happens? Animals attack first because it's their defense, and children are the same way. I don't judge, in fact, there are mothers who, when their child misbehaves, think it is because they want to. I don't judge them, but it is a process, it is not because they want to. Or when they go pee in the bathroom, and they went pee on the floor by accident, they were half asleep, and they thought it was the bathroom. I just change the sheets, and maybe have him sit in a corner, but some mothers hit the child for that, and I don't think that is correct, especially if the child is half asleep. How will they know? The child will not say, "Mommy, why did you hit me? I was so sleepy, and I didn't know what I was doing".*

Interestingly, even though this mother condemned hitting, she recognized its prevalence in society and even mentioned threatening her child by saying *boom boom*, an onomatopoeic sound that conveys the idea of hitting in Spanish.

## 4. Discussion

We sought to determine how children from low socioeconomic position in the DR are receiving the key components of the Nurturing Care Framework, particularly responsive caregiving, early opportunities for learning, and safety and security. We interviewed 25 mothers in and around the capital city to gather information about their lived experiences with these topics.

### 4.1 Summary

While most mothers did show affection to their young children, on average they underestimated or did not recognize their ability to promote their development. We found mothers provide responsive caregiving but do not view their own caregiving interactions and activities as an explicit mechanism or agent for promoting their child's learning and development. For adolescent mothers, this may be because they reported low confidence in their own parenting skills. Mothers predominantly described relying on factors external to themselves, such as tablets, to provide early learning opportunities. The reported use of physical punishment was high, particularly among adolescent mothers, which aligns with findings that over half of DR adolescents have experienced physical abuse from their parents or guardians [35].

### 4.2 Responsive caregiving

We found a divergence in findings between adolescent and adult mothers' confidence in their skills as mothers. Adolescent mothers from our sample reported greater deficits in childrearing skills compared to older mothers. The portion of adolescent mothers in our cohort is

consistent with the same proportion among the poorest quintile of mothers in the Dominican Republic (33.9%) [21]. However, while 68% in our study were under the age of 20 and 36% were under the age of 18 at first birth, all mothers in our cohort were young, with a mean age of 19.5 years and a median age of 18 years at first birth. Our findings suggest young mothers face doubt in their motherhood skills, and that those under the age of 20 experience heightened feelings of ill preparedness. Similar findings have been reported in Brazil within the context of Brazil's *Criança Feliz* early childhood program [36] and among adolescent mothers in shelters [37], and in the United States [38]. This confirms the importance of targeting this demographic for programs that promote parenting skills for child development and address socioeconomic inequities in income, access to education, and gender norms (e.g. gender-based social norms and sexual abuse), all contributors to adolescent pregnancy in the DR [39].

Improving parent awareness of how the activities they are already doing improve their child's brain development during this early stage of development, and encouraging more similar behavior could be another step toward improving the quality of stimulation children receive in the home. In particular, articulating the broad-reaching implications of critical periods in child development and how to positively influence them especially when incorporated into daily activities, such as feeding, a diaper change, a bath, or a walk to the market, could help caretakers understand the importance of their roles and increase its use in this community. Our study also suggests the importance of opportunities to deepen parents' recognition of developmental delays and ways to remediate these (i.e., improving early relational abilities in a child with aversion to touch and eye contact). Parents can benefit from access to information about stages of cognitive and language development in their children and how to recognize and promote progress in these areas for their children.

Ways to improve parent education in responsive caregiving in low-income areas have proven successful at small to medium scale, most notably through community-based home visiting programs [40]. One innovative video tool which demonstrated potential in Peru, Universal Baby, uses local footage of parent-child interactions to educate parents on brain development and responsive caregiving [41]. These studies suggest that parents would likely be receptive to additional ways to build brain power in their child (e.g., how reading aloud with a child from a book can foster attention and language; how encouraging a child to move beyond rote recitation of numbers to gain a deeper understanding of early numerical concepts such as single, few and many; or how parents can help a child build organizational and categorization skills that form the foundations of higher order thinking).

## 4.3 Intentional teaching

Few mothers took part in intentional teaching when interacting with their child, and when they did it was mostly focused on traditional recitation of letters and numbers or moral teachings such as songs and stories from church or learning the difference between right and wrong. There are very few studies investigating early learning within the Nurturing Care Framework in low- and middle-income countries, however, our results echo findings from a cross-sectional study examining knowledge and attitudes of nurturing care in a low-income cohort in Turkey, which found that knowledge and practices associated with opportunities for early learning are not strong and suggest that mothers miss critical periods for learning because they are unfamiliar with developmental milestones [42]. Likewise, one review of policy documents in Kenya found little support for early opportunities for learning among government-subsidized programs [43]. Another study among 5,570 Brazilian municipalities demonstrated the largest regional inequities in the early opportunities for learning domain, likely reflecting corresponding socioeconomic inequalities [44].

Most children this age did not attend preschool, despite INAIPI centers being available in many of their neighborhoods in Santo Domingo. This was either because they weren't aware of the program or because the program was full. Caregivers who lived in neighborhoods perceived as dangerous preferred to have their child taken care of, and sometimes even live, with a family member, often a grandmother. This may demonstrate the tension young mothers experience with competing requests for their time as students, wage earners, and childcare providers. According to the 2019 MICS report, 63% of children ages 2–4 years received some stimulating activity from a caregiver other than their parents in the three days prior to the survey, 44% participated in these activities with their mother, and 10% with their father [21]. Studies involving the nurturing care framework should include conversations with other family members, particularly grandmothers who act as primary caregivers, and explore further the role of fathers in child interactions.

Caregivers had difficulty articulating goals and dreams for their child, and how they might help them achieve these goals as caregivers. Responses centered around child behaviors such as being obedient and avoiding trouble. This reflects the structures of poverty and limited opportunities available to low-income families to help their children reach their potential. On the other hand, it also suggests that caregivers see themselves as relatively powerless in this setting. Given that child well-being is a community norm and that institutions such as school and health clinics exist in the community, as do public spaces for recreation and play, even if limited, opportunities do exist for community members to play an active role in improving opportunities and services.

Women looked primarily to external props, distal factors, and activities outside the home for learning opportunities, such as tablets and videos, saving money for their child's future education, and social interactions with other children.

Parents promotion of screens and assumptions about the educational value of screen exposure were concerning in light of growing evidence about the ways that childhood screen time may disrupt the development of healthy social skills, attention and concentration, and emotional self-regulation. The maximum recommended amount of time for a three-year-old to watch a screen is one hour per day [45,46], and excessive screen use in children ages 3–4 years is associated with lower psychosocial well-being [47]. While we did not collect data on the exact time of screentime use per day, our qualitative data suggest that most children spend several hours per day on a tablet or phone. Since 2021, the Dominican government has provided all children in elementary public schooling with a tablet, and many children in this study had access to a sibling's tablet for educational games. Some interacted with adults while playing these games, which may be less detrimental to development than unsupervised screentime [48], however, it is unclear how much other activities, such as watching videos, using social media, or surfing the internet, are conducted on the tablets. The dos Santos et al study found that caregivers use screens to calm their children during tantrums [36]. More research is needed to understand whether children in the DR are supervised while using screens, the material's quality, and adult participation in these contexts.

### 4.4 Security and safety

Although frequent in the Dominican Republic, the use of physical punishment has been shown to negatively impact child behavior and development both immediately and longitudinally [49].

We found a high use of physical punishment for disciplinary purposes, in almost half of the participants, who spontaneously reported using physical punishment at least once with their child. The 2019 MICS survey reports that an average of 8.8% of Dominican women or caregivers of young children agree with the use of physical punishment, with a higher use

among people in the lowest wealth quintile (12.6%), the lowest education level (15.0%) and among people under 25 years of age (9.3%) and over 50 years of age (9.1%) [21].

The use of physical punishment (mostly hitting with a shoe or pole) among mothers in this cohort may be augmented by the perceived helplessness and alienation that comes with adolescent motherhood. It was difficult to assess correlations with the small sample, however, 10 of the 11 mothers reporting physical punishment were under the age of 20 at first birth. These mothers reported little faith in their parenting abilities and attributed this lack of parenting ability to their own compulsive behavior. Most women who used physical punishment mentioned losing their temper and then regretting it later, while a few believed physical punishment was the best way for children to learn right from wrong.

Physical punishment is more common in low-income areas worldwide. One study in Uganda investigating attitudes about physical punishment by showing hypothetical scenes of misbehavior and asking mothers to indicate their response strategy, found that 2/3 of women report they would use physical punishment to deal with bad behavior [50]. Similarly, in a baseline assessment of women participating in a clinical trial in Afghanistan, 71.8% reported using physical punishment within the past month. Within the Brazilian sample studied by dos Santos et al, women reported using swearing and shouting to control children when they felt unable to manage difficult behavior [36], and connected this response to feeling out of control. Physical punishment can have a lasting negative impact on development. A metanalysis based on 69 surveys worldwide found that children experiencing physical punishment were on average 24% less likely to be developmentally on track [51]. Parents everywhere should have access to global data about the negative impacts that accompany corporal punishment and harsh parenting.

One objective of this paper was to deepen findings from Sánchez-Vincitore and Castro that stimulating caregiving and opportunities for early learning mediated the relationship between socioeconomic gradient and ECD outcomes among children 36–59 months of age in the Dominican Republic [24]. Our findings contribute to current knowledge of how physical punishment is used among young mothers, a population more inclined to believe physical punishment is an appropriate disciplinary technique in the MICS data, compared to older mothers. Likewise, we also show how opportunities for early learning could be improved among families in this setting. While we didn't measure the specific indicators used in Sánchez-Vincitore and Castro (2022) to predict early child development (such as number of children's books at home, stimulating activities at home, and attendance to an early childhood education), our findings suggest that the use of tablets and cell phones may be excessive and may be impeding other opportunities for learning in the home. However, among our cohort, social interaction, particularly with other children, was quite common.

## 5. Limitations

This was a small qualitative study focused on understanding how responsive caregiving, opportunities for early learning, and safety domains as recommended by the NCF are experienced by children from low-income families as reported by their mothers in and near Santo Domingo, Dominican Republic. The sample size was small; however, given the limited scope of our interviews, we reached saturation on all our topic areas of interest. We could not assess factors requiring home observation and did not cover other determinants of development, such as nutrition and childhood illness.

## 6. Conclusion

This study serves to deepen our understanding of how children from low-income households in the Dominican Republic are receiving nurturing care and to identify areas for improvement

for early intervention practitioners and public health programs. Few studies have attempted to comprehensively describe how families understand and implement nurturing care in diverse global contexts. It is vitally important as science advances our understanding of early brain biology and development that we also study the impacts of early family experience and explore the complex interplay among environmental factors such as poverty, stress, early adversity, and cultural differences in childrearing beliefs and practices.

Our findings suggest caregivers feel disempowered to teach their children and therefore seek distal tools for entertainment and education, such as tablets. This finding may curtail with our other finding that young and adolescent mothers in this setting are using physical punishment because of their own feelings of helplessness and insecurity in their motherhood skills. Both indicate mothers need more education, encouragement, and confidence to provide developmental opportunities to their children. These findings provide a clear path forward for intervention design targeting young mothers in low-income environments such as this one. Parent coaching could incorporate components of encouragement to help caregivers brainstorm creative ways to promote a sense of future for their children. In addition to building confidence and skills, future parent coaching interventions should emphasize underlying structural determinants of development, including the mother's education, socioeconomic position, gender equity, and practices that improve development particular to that setting.

Participation in enhanced institutions can serve as an incentive to help prepare children to take advantage of these services and what life chances exist. An updated report tracking global progress on the Nurturing Care Framework by the World Health Organization and UNICEF show that almost 50% of countries have increased financial support for nurturing care, including the Dominican Republic [52]. However, all institutional work must be in tandem with partnerships with community-based organizations that promote these values.

Qualitative studies are needed using ethnographic approaches to build methods to capture child caregiving from a local perspective and understand in that context what child development might mean and explore issues of access and use of ECD programs. Areas of focus could include opportunities for early learning and discussing the risks of using physical punishment for discipline. The paper by Dalava Dos Santos in 2023 found that mothers taking part in a home visiting program designed to deliver nurturing care coaching would like further coaching in positive discipline techniques, demonstrating that this is a shared issue [36].

Finally, in the Dominican Republic, it would be useful to consider cultural context by building on strengths already demonstrated in this population, such as a high level of social interaction and affectionate responsive interaction with caregivers. Families with low socioeconomic positions and adolescent mothers should be priority populations for targeted early intervention and coaching programs.

## Supporting information

**S1 File.**
(PDF)

**S2 File.**
(DOCX)

## Author Contributions

**Conceptualization:** Adrianne Katrina Nelson, Laura Sánchez-Vincitore, Michelle Marie Susana, Katherine Theall, Martha Vibbert, Heidi Luft, Arachu Castro.

**Data curation:** Adrianne Katrina Nelson, Melanie Patricia Frias, Michelle Marie Susana.

**Formal analysis:** Adrianne Katrina Nelson, Melanie Patricia Frias, Michelle Marie Susana, Arachu Castro.

**Funding acquisition:** Adrianne Katrina Nelson, Laura Sánchez-Vincitore, Arachu Castro.

**Investigation:** Adrianne Katrina Nelson, Laura Sánchez-Vincitore, Melanie Patricia Frias, Carl Kendall, Katherine Theall, Arachu Castro.

**Methodology:** Adrianne Katrina Nelson, Laura Sánchez-Vincitore, Melanie Patricia Frias, Michelle Marie Susana, Arachu Castro.

**Project administration:** Adrianne Katrina Nelson, Laura Sánchez-Vincitore, Arachu Castro.

**Resources:** Adrianne Katrina Nelson, Laura Sánchez-Vincitore, Arachu Castro.

**Software:** Adrianne Katrina Nelson, Arachu Castro.

**Supervision:** Adrianne Katrina Nelson, Laura Sánchez-Vincitore, Carl Kendall, Katherine Theall, Martha Vibbert, Heidi Luft, Arachu Castro.

**Validation:** Adrianne Katrina Nelson, Laura Sánchez-Vincitore, Melanie Patricia Frias, Carl Kendall, Martha Vibbert, Heidi Luft, Arachu Castro.

**Visualization:** Adrianne Katrina Nelson, Melanie Patricia Frias, Carl Kendall, Heidi Luft, Arachu Castro.

**Writing – original draft:** Adrianne Katrina Nelson, Martha Vibbert, Heidi Luft.

**Writing – review & editing:** Adrianne Katrina Nelson, Laura Sánchez-Vincitore, Melanie Patricia Frias, Michelle Marie Susana, Carl Kendall, Katherine Theall, Martha Vibbert, Heidi Luft, Arachu Castro.

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
