## [Decision Letter · Decision Letter 0]

4 Jun 2024

PONE-D-23-41977Home-based nurturing care for children under five with low socioeconomic position in Santo Domingo, Dominican RepublicPLOS ONE

Dear Dr. Nelson,

Thank you for submitting your manuscript to PLOS ONE. After careful consideration, we feel that it has merit but does not fully meet PLOS ONE’s publication criteria as it currently stands. Therefore, we invite you to submit a revised version of the manuscript that addresses the points raised during the review process.

We look forward to receiving your revised manuscript.

Kind regards,

Mohammad Nayeem Hasan

Academic Editor

PLOS ONE

3. In the online submission form, you indicated that [Data cannot be shared publicly because of they are identifiable qualitative data. Data are available upon request from the corresponding author for researchers who meet the criteria for access to confidential data.]. 

Reviewers' comments:

Reviewer's Responses to Questions

**Comments to the Author**

1. Is the manuscript technically sound, and do the data support the conclusions?

Reviewer #1: Yes

Reviewer #2: Yes

Reviewer #3: Yes

2. Has the statistical analysis been performed appropriately and rigorously? 

Reviewer #1: Yes

Reviewer #2: N/A

Reviewer #3: I Don't Know

3. Have the authors made all data underlying the findings in their manuscript fully available?

Reviewer #1: No

Reviewer #2: Yes

Reviewer #3: No

4. Is the manuscript presented in an intelligible fashion and written in standard English?

Reviewer #1: Yes

Reviewer #2: Yes

Reviewer #3: Yes

5. Review Comments to the Author

Reviewer #1: A very well versed and detailed article and research conducted in a developing country. Nurturing framework is very pivotal and much needee glovally such kind of studies support the widespread implementation of ECD and NCF. The study uniquely highlighted intentional learning and corporal punishment.However following are the points of feedback:

The title needs to be modify as research question and study is about perceptions of mothers in DR on NCF, weather they provide such care or not however title suggest home basee NCF as a intervention in homes)

Introdution (1.1, 1.2, 1.3 can be combine into one short paragraph)

1.6 ( Add DR country profile overall, what is the status of ECD strategies, why this country was selected for study, why there is high prevalence 14.5%, what is the current situation of government interventions?)

Discussion: Adding other developing countries data literture to support the study.

Conclusion: Adding recommendations as to government role and policies, input of WHO and other NGOs

Reviewer #2: The author cannot explains how the empirical

findings contribute to the broader literature; discussing on how the

findings address a gap in the literature, extend our current knowledge, or

provide new insights into a specific phenomenon or problem

Reviewer #3: Congratulation on your works and the writing. My comment as submitted on the PLOS ONE platform. Please refer to the comments and please revised it before you resubmit the manuscript especially using of the right term in scientific writing. All the best and good luck

6. PLOS authors have the option to publish the peer review history of their article (what does this mean?). If published, this will include your full peer review and any attached files.

Reviewer #1: No

Reviewer #2: No

Reviewer #3: **Yes: **Khairul Hasnan Amali

---

## [Author Response · Author response to Decision Letter 0]

23 Aug 2024

PONE-D-23-41977

Response to reviewers

We have done this.

We include the questionnaire here.

3. In the online submission form, you indicated that [Data cannot be shared publicly because of they are identifiable qualitative data. Data are available upon request from the corresponding author for researchers who meet the criteria for access to confidential data.]. 

Following the ethics committee protocol approved by Tulane University (2019-2375), the audio files and transcripts, which contain sensitive information and are provided in conditions of confidentiality, cannot be shared. This follows the recommendations made by Martijn de Koning and colleagues in "Guidelines for anthropological research: Data 

management, ethics, and integrity" (Ethnography, 2019. 20(2): p. 170-174).

I can provide de-identified qualitative data coded and converted to numerical form.

We have included this citation in the clean version of the manuscript.

Response to Reviewers' Comments:

1. Is the manuscript technically sound, and do the data support the conclusions?

Reviewer #1: Yes

Reviewer #2: Yes

Reviewer #3: Yes

2. Has the statistical analysis been performed appropriately and rigorously? 

Reviewer #1: Yes

Reviewer #2: N/A

Reviewer #3: I Don't Know

3. Have the authors made all data underlying the findings in their manuscript fully available?

Reviewer #1: No

Reviewer #2: Yes

Reviewer #3: No

4. Is the manuscript presented in an intelligible fashion and written in standard English?

Reviewer #1: Yes

Reviewer #2: Yes

Reviewer #3: Yes

Review Comments to the Author

Reviewer #1: A very well versed and detailed article and research conducted in a developing country. Nurturing framework is very pivotal and much needed globally such kind of studies support the widespread implementation of ECD and NCF. The study uniquely highlighted intentional learning and corporal punishment. However, following are the points of feedback:

The title needs to be modified as research question and study is about perceptions of mothers in DR on NCF, weather they provide such care or not however title suggest home based NCF as an intervention in homes).

Thank you for this observation. We have changed the title to: Home-based nurturing care practices for children under five with low socioeconomic position in Santo Domingo, Dominican Republic

Introduction (1.1, 1.2, 1.3 can be combine into one short paragraph).

We have combined these subsections into one sub-section entitled “The Nurturing Care Framework”, however, we did not make it into one paragraph as that would be very long.

1.6 (Add DR country profile overall, what is the status of ECD strategies, why this country was selected for study, why there is high prevalence 14.5%, what is the current situation of government interventions?)

Thank you for this comment. We have added in a sentence about why the DR was chosen for this study. There are few studies on neurodevelopment in the DR and so we cannot point to evidence as to why the prevalence of delay is high beyond the reasons we have offered (high rates of poverty, low levels of early learning opportunities). The second paragraph of section 1.6 (now 1.4) describes available government interventions. We do not have access to evaluation data on INAIPI but hope that will become available within the next few years. 

Discussion: Adding other developing countries data literature to support the study.

Thank you for this comment. We agree and have added in findings from other low-income countries about responsive caregiving education, physical punishment, and the use of screens and well as information from systematic reviews and metanalyses.

Conclusion: Adding recommendations as to government role and policies, input of WHO and other NGOs.

Thank you, we have added in some language about strengthening institutions and WHO recommendations into the conclusion.

Reviewer #2: The author cannot explain how the empirical findings contribute to the broader literature; discussing on how the findings address a gap in the literature, extend our current knowledge, or provide new insights into a specific phenomenon or problem.

The authors thank you for this comment. We have included additional literature on the primary topic areas we contribute to the knowledge in the field in the discussion section. We have also included a summary paragraph in the conclusion section highlighting key take-aways. Our findings suggest caregivers are not empowered to teach their children and therefore seek distal tools for entertainment and education. This may curtail with our findings that young and adolescent mothers in this setting are using physical punishment because of their own feelings of helplessness and insecurity in their motherhood skills. These findings provide a clear path forward to build confidence and knowledge among young mothers. Additionally, we provide evidence that ethnographic research about caregiving practices may facilitate promoting nurturing care from local experiences. 

Reviewer #3: Congratulation on your work and the writing. My comment is submitted on the PLOS ONE platform. Please refer to the comments and please revise it before you resubmit the manuscript, especially using the right term in scientific writing. All the best and good luck.

I have been unable to view these suggested revisions and we’re looking forward to incorporating them.

---

## [Decision Letter · Decision Letter 1]

29 Oct 2024

PONE-D-23-41977R1Home-based nurturing care practices for children under five with low socioeconomic position in Santo Domingo, Dominican RepublicPLOS ONE

Dear Dr. Nelson,

Thank you for submitting your manuscript to PLOS ONE. After careful consideration, we feel that it has merit but does not fully meet PLOS ONE’s publication criteria as it currently stands. Therefore, we invite you to submit a revised version of the manuscript that addresses the points raised during the review process. Addressing the following issues benefit  your manuscript, in addition to the reviewers comments.It appears that 'Table 2. Select excerpts' could be better organised for clarity.It is better to include protocol numbers in the ethics approval statement (i.e., 'We received approval from the Universidad Iberoamericana Human Research Protection Program and the Tulane University Research Ethics Committee').Although detailed quotes provide context, some are lengthy, particularly in lines 596-611 and 636-655. Given the abundance of quotes in Table 2, consider condensing them to improve readability and avoid excess detail.In your submission, there are two clean copies and one trach change. Could you please do only one clean copy and one track change?

Please submit your revised manuscript by Dec 13 2024 11:59PM. If you will need more time than this to complete your revisions, please reply to this message or contact the journal office at plosone@plos.org. Please include the following items when submitting your revised manuscript:

We look forward to receiving your revised manuscript.

Kind regards,

Yitagesu Habtu Aweke, Ph.D

Academic Editor

PLOS ONE

---

## [Author Response · Author response to Decision Letter 1]

8 Nov 2024

Response to Review 2

11/7/24

• It appears that 'Table 2. Select excerpts' could be better organised for clarity.

Thank you for noticing this. The subthemes are in the same order they are presented in the paper. However, we re-read the citations and removed/ moved some to make sure each was consistent with its theme. We also added a short summary of the theme for each.

• It is better to include protocol numbers in the ethics approval statement (i.e., 'We received approval from the Universidad Iberoamericana Human Research Protection Program and the Tulane University Research Ethics Committee').

Yes, we have added the protocol numbers into this version.

• Although detailed quotes provide context, some are lengthy, particularly in lines 596-611 and 636-655. Given the abundance of quotes in Table 2, consider condensing them to improve readability and avoid excess detail.

We have condensed these citations as suggested.

• In your submission, there are two clean copies and one trach change. Could you please do only one clean copy and one-track change?

Yes, please forgive this oversight. We have corrected this in the current submission.

---

## [Editor Report · Decision Letter 2]

11 Nov 2024

Home-based nurturing care practices for children under five with low socioeconomic position in Santo Domingo, Dominican Republic

PONE-D-23-41977R2

Dear Dr  Adrianne Katrina Nelson,

We’re pleased to inform you that your manuscript has been judged scientifically suitable for publication and will be formally accepted for publication once it meets all outstanding technical requirements.

Kind regards,

Yitagesu Habtu Aweke, Ph.D

Academic Editor

PLOS ONE

---

## [Editor Report · Acceptance letter]

19 Nov 2024

PONE-D-23-41977R2 

PLOS ONE

Dear Dr. Nelson, 

I'm pleased to inform you that your manuscript has been deemed suitable for publication in PLOS ONE. Congratulations! Your manuscript is now being handed over to our production team.

Kind regards, 

on behalf of

PhD Candidate Yitagesu Habtu Aweke 

Academic Editor

PLOS ONE